# The Fröhlich-Morchio-Strocchi mechanism and quantum gravity

**Axel Maas**

Institute of Physics, NAWI Graz, University of Graz,
Universitätsplatz 5, A-8010 Graz, Austria

## Abstract

Taking manifest invariance under both gauge symmetry and diffeomorphisms as a guiding principle physical objects are constructed for Yang-Mills-Higgs theory coupled to quantum gravity. These objects are entirely classified by quantum numbers defined in the tangent space. Applying the Fröhlich-Morchio-Strocchi mechanism to these objects reveals that they coincide with ordinary correlation functions in quantum-field theory, if quantum fluctuations of gravity and curvature become small. Taking these descriptions literally exhibits how quantum gravity fields need to dress quantum fields to create physical objects, i. e. giving a graviton component to ordinary observed particles. The same mechanism provides access to the physical spectrum of pure gravitational degrees of freedom.

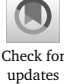

# 1 Introduction

In non-gravitational quantum field theories, global and local symmetries play fundamentally different roles [1–4]. Local symmetries localize theories, and are essentially auxiliary. This can probably be best seen from the fact that they can be removed by a choice of suitable variables, leaving theories having only (almost) global symmetries [3–8]. This shows that in principle physics should not depend on the treatment of local symmetries, especially not on any gauge choices. Though in practice this is not a very useful insight, especially due to the Gribov-Singer ambiguity [3, 9, 10], it is important conceptually. Taken to the extreme, this implies that effects like confinement, in the sense of the absence of colored states from the spectrum, are nothing but a manifestation that they are gauge-dependent, and hence unphysical [3].

Global symmetries, on the other hand, have important observable consequences [4, 5]. While global charges are, strictly speaking, also not directly observable, differences in global charges are. Especially, this leads to physical effects like degeneracies, super selection sectors, and allowed and forbidden decays. In this sense, global symmetries are physical.

These insights can be used to formulate observables, e. g. in the standard model [1, 2, 4], in a manifest gauge-invariant way. Interestingly, the gauge-dependent degrees of freedom encode still information on the gauge-invariant physics, which can be used to reconstruct observable physics. While this in general requires non-perturbative methods, in many cases quantum fluctuations are small compared to classical physics, allowing a systematic treatment. This essentially reduces to expanding around a classical solution. E. g., in QED [3] this is done around the quantum-mechanical exact solution of the hydrogen atom. In electroweak physics, this is possible due to the Fröhlich-Morchio-Strocchi (FMS) mechanism [1, 2], which expands gauge-invariant states around vacuum expectation values. It is this latter approach, which will also be useful here. A review of the FMS mechanism in particle physics can be found in [4].

Of course, an immediate question is, what happens in a situation with gravity. In that case coordinate transformations become local themselves, and thus the line between global and local symmetries seems to blur. This seems to be especially true for quantum fields on a fixed singular background metric, e. g. a Schwarzschild or Kerr metric. However, a full treatment requires quantum gravity, to put all entities on the same footing.

The aim of the present work is to construct a manifestly invariant description of objects in quantum gravity, which yield the usual particles of quantum field theory if the quantum gravitational fluctuations become negligible. This will also help in defining the role both of global and local symmetries in the quantum gravity setup. Employing the FMS mechanism will also allow to obtain an approximate calculation scheme to obtain their properties.

Of course, this depends on the setup for quantum gravity. Here, canonical quantum gravity will be used, under the assumption that it becomes well-defined in a path-integral approach, e. g. due to asymptotic safety [11–13] or some other non-perturbative mechanism [14, 15]. An important role will be played by the tangent space, which will be needed to give global symmetries a well-defined meaning, which leads to a somewhat unusual perspective. However, it also allows to make connections to other, more general, approaches to gravity, like Kibble-Sciama [16–18] or spin basis [19, 20] ones. The details of the setup will be discussed in sections 2 and 3. This allows for both well-defined spin quantum numbers and will act as a guiding principle to construct global symmetries of particles in quantum gravity.

In section 4 first the emergence of conventional particles in this setup will be studied using the FMS mechanism. To consider both global and local symmetries the simplest example is Yang-Mills-Higgs theory, in the version which forms the electroweak sector of the standard model. Adding fermions to the mix would substantially complicate the issue of spin [18, 20], and will not be considered here. Indeed, it will be seen how the physical spectrum reemerges when the particles are studied in situations where quantum gravity effects are small. At the

same time, a constructive way will be obtained how to add both curvature effects as well as quantum fluctuations of gravity.

A natural consequence of requiring manifest invariance requires to discuss how a graviton emerges. In preparation for this, first the simpler case of scalar geons [21, 22] will be discussed in section 4. To investigate the graviton requires spin, as otherwise it is not possible to distinguish geons and gravitons, and also particle physics excitations, as will be discussed in section 5.

Taking the presented results at face value leads to interesting speculation for their consequences for phenomenology, which will be done in section 6. This poses interesting questions, which will require non-perturbative techniques to fully answer. Simulations in the spirit of [23–25] are most likely suited to directly answer these questions as well as functional methods [11–13, 26, 27], but this is also possible using other approaches [14, 15]. These future directions will be discussed in the summary in section 7.

## 2  Setup

The basic setup in the following will be to consider (four-dimensional) space-time as a collection of events, which can be enumerated, and have definite neighboring relations[1]. To every event is associated a flat, Minkowski tangent space, and a vierbein field $e_\mu^a$ [28], which connects the tangent spaces to the manifold of coordinates in the usual way, i. e. by mapping the corresponding unit vectors $E$ into each other, $E_\mu = e_\mu^a E_a$, where Greek indices count in the manifold and Latin indices in the tangent space. The vierbein is required to be locally invertible, making this a bidirectional connection.

Moreover, a metric on the manifold can then be constructed as

$$g_{\mu\nu} = e_\mu^a e_\nu^b \eta_{ab},\tag{1}$$

where $\eta_{ab}$ is the flat Minkowski metric in the tangent space. From the metric the usual Christoffel symbols $\Gamma^\nu_{\mu\rho}$ and Christoffel symbols with mixed indices can be constructed by using the vierbein. Especially, this defines the spin connection

$$\Gamma^\mu_{ab} = e_a^\rho e_b^\sigma \Gamma^\mu_{\rho\sigma}.\tag{2}$$

The covariant derivative then takes the form

$$D_\mu = e_\mu^a D_a = \partial_\mu + \Gamma^{ab}_\mu f_{ba},\tag{3}$$

where $f_{ba}$ are the usual generators of the Lorentz group in the corresponding representation of the object in question.

The metric is assumed compatible, i. e. $D_\mu g_{\rho\sigma} = 0$, what ensures that the causal connection between neighboring events is the same, no matter what is the starting event. Finally, this metric is taken to be torsion free, and thus

$$\Gamma^\rho_{\mu\nu} = \frac{1}{2}g^{\rho\sigma}\left(\partial_\mu g_{\nu\sigma} + \partial_\nu g_{\mu\sigma} - \partial_\sigma g_{\mu\nu}\right)\tag{4}$$

as usual.

Under a general coordinate transformation in the manifold $x_\mu \to x_\mu + \xi_\mu(x_\mu) = J^\nu_\mu x_\nu$ the vierbein transforms homogeneously $e \to Je$ and so does the spin connection $\Gamma \to J\Gamma$. Such a

---

[1]When thinking of the underlying $\mathbb{R}^4$ which is used to define a manifold [28], this $\mathbb{R}^4$ could be used to enumerate the events and define neighboring relations.

transformation acts on the manifold indices $\mu$. It should be noted that rotations and Lorentz boosts are also represented by such event-dependent translations [18, 29]. Especially, orbital angular momentum is connected to these local translations.

To also include spin, it is necessary to include a Lorentz group acting on the tangent space, but not the manifold [18, 29]. Under this transformation the vierbein transforms also homogeneously $e \to \Lambda e$. Because of that, however, this is actually immediately a local transformation, if the dynamics depend only on the metric, like in the Einstein-Hilbert action (8) below: Any local Lorentz transformation drops out immediately in (1). Especially, also the Christoffel symbols (4) are trivially invariant under it. This is not true for the spin connection (2), which transforms as

$$\Gamma_\mu \to \Lambda \Gamma_\mu \Lambda^{-1} + \Lambda D_\mu \Lambda^{-1}, \tag{5}$$

with the covariant derivative (3).

It should be noted that both transformations do not commute [18]. By introducing the covariant derivative as (3), spin is defined in the Lorentz representations of the tangent space. This assignment does not mix spin with orbital angular momentum, which is defined in the manifold [18]. Thus, tensors with tangent indices transform like Lorentz tensors in particle physics, and can be associated to have a fixed spin, which is essentially their representation of the Lorentz group. The local part of the Lorentz symmetry acts then merely as a trivial reparametrization symmetry. This will become more complicated once fermions are introduced, but this will not be investigated here.

If one would want to make the Lorentz symmetry also dynamical in the sense of a gauge symmetry, this would require to introduce additional fields. One possibility would be to make the spin connection (2) such an independent field [16–18], another one using dynamical Dirac matrices [19, 20]. In general, this will allow for space-time torsion. These options will not be considered here. In fact, in section 5 it will be seen that such local Lorentz symmetry is actually potentially problematic at the quantum level, if one would like to keep spin as an observable.

From these objects the Riemann tensor can be constructed as [18]

$$R_{\mu\nu\rho}{}^{\sigma} = e^a_\rho e^\sigma_b \eta_{ac} F_{\mu\nu}{}^{cb} \tag{6}$$

$$F_{\mu\nu}{}^{ab} = 2\left(\partial_{[\mu}\Gamma_{\nu]}^{ab} + \eta_{cd}\Gamma_{[\mu}^{ca}\Gamma_{\nu]}^{db}\right), \tag{7}$$

and the corresponding contractions create the Ricci tensor and curvature scalar. A suitable classical action for this theory can be constructed as [18]

$$S = \frac{1}{2\kappa}\int d^4x \det(e)\left(e^\mu_a e^\nu_b F_{\mu\nu}{}^{ab} + l\right). \tag{8}$$

Here, $\kappa$ and $l$ are the usual combinations of Newton's constant and the cosmological constant. The experience [11–15, 24, 25] strongly suggests that this action is insufficient at the quantum level, and that higher-order terms, e. g. of $R^2$ and spin type [18], are necessary. As all calculations here will remain at lowest-order tree-level, this does not need to be specified yet, though the quantitative results below may, of course, change.

In the following this theory will be coupled to Yang-Mills-Higgs theory, i. e. the weak/Higgs sector of the standard model. For the scalar the coupling to the spin connection vanishes automatically in the Lagrangian, as for the trivial representation $f_{ab} = 0$. It is assumed that there is also no coupling to the gauge field, as otherwise gravity already classically breaks the gauge symmetry [18, 28]. Especially, in such a case this corresponds to a gauge anomaly, i. e. the quantum theory would depend on the choice of gauge in the classical theory[2]. Conversely,

---

[2] It may be that this would actually be possible, see e. g. [30], but this will not be considered here.

the angular momentum current would change under a gauge transformation, which would have already at weak gravity consequences.

Hence, the weak-gauge covariant derivative $\Delta$ remains covariant only with respect to the gauge field. This yields the matter action as

$$S_m \quad = \quad \int d^4x \det(e)\Big(g^{\mu\nu}(\Delta_\mu^{pq}\phi_u^q)^\dagger(\Delta_\nu^{pr}\phi_u^r) + V(\phi^\dagger\phi) + g^{\mu\rho}g^{\nu\sigma}W_{\rho\sigma}^i W_{\mu\nu}^i\Big) \tag{9}$$

$$\Delta_\mu^{pq} \quad = \quad \partial_\mu\delta^{pq} - igW_\mu^i T_{pq}^i \tag{10}$$

$$W_{\mu\nu}^i \quad = \quad \partial_\mu W_\nu^i - \partial_\nu W_\mu^i - gf^{ijk}W_\mu^j W_\nu^k, \tag{11}$$

where letters $i,\dots$ enumerate the adjoint representation of the gauge algebra, $p,\dots$ the fundamental representation of the gauge algebra, and $u,\dots$ the fundamental representation of the custodial symmetry, i. e. the flavor symmetry of the Higgs degrees of freedom [4], both assumed to be an SU(2) group, in accordance with the standard model.

This theory features covariantly conserved angular momentum, which is entirely made from orbital angular momentum, as the spin of the gauge fields is not coupling to the spin connection. Energy and momentum is covariantly conserved, in the usual sense [18, 28].

The action (11) is trivially invariant under custodial transformations, which are performed event-independent. In this sense, the symmetry remains global. In effect, custodial transformations commute with space-time symmetries. The corresponding Noether current

$$J_\mu^u \quad = \quad \Im\mathrm{tr}\Big(T^u X^\dagger\Delta_\mu X\Big) \tag{12}$$

$$X \quad = \quad \begin{pmatrix} \phi_1 & -\phi_2^\dagger \\ \phi_2 & \phi_1^\dagger \end{pmatrix}, \tag{13}$$

with $T^u$ the generators of the custodial symmetry, is covariantly conserved

$$D^\mu J_\mu^u = \partial^\mu J_\mu^u = 0 \tag{14}$$

as none of the building blocks couple to the spin connection, the coupling constants $f$ in (3) are all zero [18], and thus the conservation reduces to the ordinary one. This is quite similar to the electromagnetic case [18], where the covariant derivative in the current conservation can also be replaced by an ordinary one. It is precisely due to the absence of a coupling of the gauge fields to the spin connection and the fact that the charge carriers are scalars that this is possible.

To summarize, the theory has four different symmetries. One is the diffeomorphism on the manifold, which takes the form of a local transformation of the vierbein or metric as the elementary degree of freedom. There is the Lorentz transformation, which acts at every event only on the vierbein, but trivially on the action. Nonetheless, the diffeomorphisms and the Lorentz transformation do not commute when applied to the vierbein. There is a global custodial transformation $C$, which acts only on the Higgs field, in the same way at every event. Finally, there is a gauge transformation $G$. It transforms both the $W$ field and the Higgs field in the internal space at every event in a local way. In this context, the gauge-field acts like the connection. Thus, under transformations $J$, $C$, $G$, and $\Lambda$ the independent fields behave as

$$e_\mu^a \quad \rightarrow \quad J_\mu^\nu(x)\Lambda_b^a e_\nu^b \tag{15}$$

$$X_{pu} \quad \rightarrow \quad G_p^q(x)C_u^\nu X_{q\nu} \tag{16}$$

$$W_\mu^{pq} \quad \rightarrow \quad J_\mu^\nu(x)G_r^p(x)G_s^{-1q}(x)W_\nu^{rs} + J_\mu^\nu(x)G_r^p\big(\partial_\nu G^{-1}\big)^{qr}, \tag{17}$$

where $W_{pq}^\mu = W_i^\mu T_{pq}^i$ and the change of evaluation coordinates $x$ of the events has been suppressed for brevity.

# 3 The quantum theory

To obtain a quantum version of this theory requires at the moment various assumptions. It will here be assumed that a path integral formulation

$$Z = \int \mathcal{D}e\mathcal{D}\phi\mathcal{D}W e^{i(S+S_m)} \tag{18}$$

works. Note that in a sense of a geometrical definition the fields are functions of events, i. e. every field configuration gives a field amplitude at every event. Only to actually calculate a quantity like the action coordinates need to be introduced.

Herein are made three central assumptions. One is that the vierbein is the suitable dynamical variable. However, exchanging it for the metric would lead to essentially no change in the remainder. The motivation to chose the vierbein instead of the metric is to have a dynamical variable carrying the information about spin. The second is that the measure is a suitable Haar measure without adding further terms to the action to avoid possible obstructions [31]. And finally that it is necessary to integrate over the full, non-compact GL(4,$\mathbb{R}$) group, i. e. without restrictions of the possible manifolds. Especially, it is assumed that diffeomorphism orbits have all the same (infinite) size.

Note that the measure is also invariant under the local Lorentz transformations and diffeomorphisms. However, local Lorentz symmetry enters in a trivial way. Especially, the partition sum is well defined without fixing this symmetry, and its volume could be absorbed in the normalization. This is in contrast to the diffeomorphism symmetry, which would need treatment before perturbative expressions could be calculated.

That these assumptions are probably insufficient beyond tree-level is shown by the apparent necessity of counter terms and probable need for extended gravity actions [14, 15, 24, 25]. Also, at loop-order a dynamical effect like, e. g., asymptotic safety [11–13, 26] will be needed to make the theory well-defined. All of this will be necessary when pushing the present investigations beyond the tree-level ones to follow. However, as it will be seen, this may be only a relatively small quantitative effect except at the most extreme of situations.

There is one important consequence of (18) which is true for it and for any of its extension which does not introduce absolute frames: Just as with quantum field theories [1, 2, 4, 32], this implies that quantities not invariant under local diffeomorphism transformations, or gauge transformations, have necessarily zero expectation value[3], as long as no coordinate system is fixed. The reason is that the path integral (18) sums over all possible values of the vierbein, and thus metric, with equal weight, as the action and measure are invariant. Thus, for every non-invariant quantity the path integral is like integrating over a sphere, leaving no net direction. The implicit assumption is, as with gauge theories, that all diffeomorphism orbits have the same (infinite) size. Then an evaluation of (18) is well-defined, as this can be recast in a sum over all orbits separately. In practice, because GL(4,$\mathbb{R}$) is a non-Abelian group, this may be involved due to the Gribov-Singer ambiguity [9, 10].

Especially, this implies that $\langle g_{\mu\nu} \rangle = 0$. Likewise, any quantity carrying non-contracted indices, no matter whether tangential ones, space-time ones, or belonging to the custodial or gauge symmetry, has necessarily vanishing vacuum expectation values. Just because every possible transformation of it will be integrated over as well. Thus, the only non-vanishing vacuum expectation values are those with fully contracted indices, which are also invariant under local transformations. Thus, they can only be products of operators $\mathcal{O}^{a\cdots r\cdots}(x)$, which

---

[3]Note [4] that an individual measurement will very much yield a non-zero value - provided the measurement process remains in quantum gravity as it is in quantum mechanics.

are diffeomorphism and gauge invariant, and are contracted as[4]

$$\omega_{a_1 a_2 \dots r_1 r_2 \dots} \mathcal{O}^{a_1 \dots r_1 \dots}(x) \mathcal{O}'^{a_2 \dots r_2 \dots}(y) \dots \tag{19}$$

Herein is $\omega$ a constant tensor, built as a tensor product from arbitrary invariant tensors of all involved groups of suitable rank. As a consequence, this requires to construct objects which transform in a suitable way under all symmetries to form physical observables.

The simplest example are operators, which are completely scalar, and thus invariant. Two very interesting such scalar operators are

$$O_1(x) = \phi^\dagger(x)\phi(x), \tag{20}$$

which describes the physical Higgs particle [4], and

$$O_2(x) = R(x), \tag{21}$$

the local curvature. They will play an important role later one. It should be noted that also operators like (20) and (21) depend on the event $x$, not on coordinates.

In particular, an operator like (21) would also allow to characterize the average space-time structure, despite $\langle g_{\mu\nu} \rangle = 0$, by forming the expectation value

$$\frac{\left\langle \int d^4 x \det(e) R(x) \right\rangle}{\left\langle \int d^4 x \det(e) \right\rangle}. \tag{22}$$

If no event is special, the average space-time needs to be necessarily homogeneous and isotropic, and thus this invariant average curvature is sufficient to characterize it. If it is suspected that the average space-time has a more involved structure, this is not trivial. One possibility is to gauge-fix diffeomorphism invariance and the coordinate system, which allows to give $\langle g_{\mu\nu}(x) \rangle$ again a meaning. Alternatives would be to determine distributions of $ds^2$, which are also invariant quantities[5].

The interesting question is now how to associate conventional particle physics objects in a physical sense to operators like (20) and (21). The simplest physical object in particle physics is the particle itself. While the notion of particle in itself is quite non-trivial [5], the fundamental quantity describing it is less ambiguous: The propagator. Any resemblance to physical propagators will require a dependence on two events, e. g.

$$D(x, y) = \langle O(y) O(x) \rangle. \tag{23}$$

The points $x$ and $y$ are taken to denote the events, not the coordinates. Hence, at this point, the propagator is not a function of distance, but of two events. This dependence is coordinate-independent, and thus the whole expression is diffeomorphism-invariant. The dependence on the events can be exchanged for any diffeomorphism-invariant characterization of the two events.

In flat-space quantum field theory this is actually also true. But because space-time is static, the events are in a one-to-one correlation with coordinates, and thus it is not obvious. The characterization of the two events then becomes just the ordinary distance, which is static as well, giving the usual notion of a propagator depending on the distance between two points.

---

[4]Similar to particle physics [3, 5, 8], it could be possible to attain gauge invariance by a suitable dressing in the spirit of the Dirac string, see e. g. [33, 34]. The present approach has a very similar relation to this as in the particle physics case [4].

[5]The same statements should apply if the equations of motions of the quantum effective action are solved. Especially, for a non-gauge fixed action without explicit coordinate system I would still expect $\langle g_{\mu\nu} \rangle$ vanishes [1, 2].

In the quantum gravity space-time is expected to be isotropic on average as well. Hence, again only one quantity is needed for a diffeomorphism-invariant characterization of the relation between both events. However, this is more complicated now, as distance depends on the metric, and is thus dynamical as well. Thus, the distance between two events becomes itself an expectation value [23]. However, for every configuration there is a unique geodesic connecting the two events $x$ and $y$ [28]. Thus, a uniquely defined expectation value for an invariant length $r$ can be defined as

$$r(x,y) = \left\langle \min_{z(t)} \int_x^y dt\, g^{\mu\nu} \frac{dz_\mu(t)}{dt} \frac{dz_\nu(t)}{dt} \right\rangle. \tag{24}$$

In this the minimization over the path $z(t)$ connecting the events $x$ and $y$ should state to find the geodesic length. With this, the propagator (23) should be considered to be a function of the expectation value of this geodesic distance, $D(r(x,y))$. This creates an invariant under all gauge symmetries, both local and space-time, and is hence a physical object. Of course, to calculate (23) and (24) it is usually necessary to introduce again coordinates. But provided the calculational scheme preserves diffeomorphism invariance the result will be independent of this.

As will now be seen, this definition recovers in the quantum-field theoretical limit systematically the ordinary propagator. Especially, with the space-time being static in flat-space-time quantum field theory, the second expectation value (24) becomes trivial, and can therefore always be performed implicitly.

## 4 The FMS mechanism and emergence of flat-space-time quantum field theory

That the expectation value of the vierbeins, and of the metric, vanishes is a consequence of diffeomorphism invariance and the path integral averaging over the whole group. Classically, of course, the Einstein equations have only solutions with $g_{\mu\nu} \neq 0$.

As the diffeomorphism invariance in (18) behaves like a gauge symmetry, it is possible to fix a gauge. This is essentially equivalent to fixing a coordinate system. As in gauge theories, this can be implemented by inserting a functional $\delta$-function in such a way as to only pick up the contribution of a single representative of every diffeomorphism orbit. This will not alter the value of any diffeomorphism-invariant quantity, though diffeomorphism-dependent quantities, like the metric, will change, and depend on the choice. Because of the non-Abelian structure, this procedure could [35–37] suffer from a Gribov-Singer ambiguity [9, 10]. However, the consequence of this will make the argument of the $\delta$-function just (much) more involved. But aside from this technical complication this will have no impact on the conceptual development here.

It is possible to fix any coordinate system. Especially, it can be fixed such that the gauge-fixed vacuum expectation value of the metric no longer vanishes. In this sense, it is very similar to what happens in Brout-Englert-Higgs physics, where the vacuum expectation value of the Higgs only appears in some fixed gauges, but vanishes in other gauges or without gauge fixing [1, 2, 4, 38]. Furthermore, all gauge-invariant quantities remain untouched. E. g., the curvature, as an invariant quantity, has still the same value.

The advantage is that there may exist gauges in which calculations become especially easy. E. g., just as the observed Fermi constant in a gauge with non-vanishing Higgs vacuum expectation value can be given a simple form and calculated at tree-level, so it is may be possible to construct gauges with a simple connection to one or more observables also in quantum gravity.

Of course, the choice of such gauges will depend in general on the values of the parameters of the theory, i. e. Newton's constant and the cosmological constant.

Consider, e. g., a situation where the observed curvature is that of a de Sitter vacuum[6]. Then it is possible to fix a gauge such that the vacuum expectation value of the metric is the de Sitter metric, and just as in Brout-Englert-Higgs physics it is possible to split[7]

$$g_{\mu\nu} = g^c_{\mu\nu} + \gamma_{\mu\nu}, \tag{25}$$

where $g^c_{\mu\nu}$ is the classical de Sitter solution, and $\gamma_{\mu\nu}$ are the quantum fluctuations satisfying $\langle \gamma_{\mu\nu} \rangle = 0$. Such a choice will be called curvature gauge in the following. Note that only $g_{\mu\nu}$ and $g^c_{\mu\nu}$ are separately necessarily genuine metrics, but $\gamma_{\mu\nu}$ may not be. However, because $\gamma_{\mu\nu}$ will be mainly small in the following, this will not be an issue at leading order.

This choice is, of course, not necessary, only convenient. Another possible split could be as well $g_{\mu\nu} = \eta_{\mu\nu} + \gamma'_{\mu\nu}$, but then the curvature would be entirely created from the quantum fluctuations, and not from the splitted classical part. Because the calculation of the curvature would then be hard, this would not be a good choice.

The FMS idea [1, 2, 4] is to take an operator, and insert the split (25) into its expression. This is merely an exact rewriting, which does not necessarily have any useful consequences. However, in BEH physics, the usefulness comes from the fact that physical quantities are dominated by the classical part, because the average amplitude of quantum fluctuations of the Higgs field are small compared to the vacuum expectation value. Then, the FMS mechanism yields a way of determining physical non-trivial results [1, 2, 4]. If now in the same sense the fluctuations around the classical metric are small, then this can also be used in quantum gravity in the same way. As the universe around us shows, just like for BEH physics, exactly such a behavior this seems to be not so a bad starting point. Of course, this is a coincidence due to the parameter values of gravity in our universe, just as is the case for BEH physics.

Implementing this idea implies to expand any correlation function of an operator $O(g_{\mu\nu})$, in which the metric $g_{\mu\nu}$ appears $n$ times, as

$$\langle O(g_{\mu\nu}) \rangle = \langle O(g^c_{\mu\nu}) \rangle + \sum_{i, \text{Permutations of } g^c \text{ and } \gamma} \langle O(g^c_{\mu\nu}; n-i, \gamma_{\mu\nu}; i) \rangle, \tag{26}$$

where the numbers in the second argument indicate how often the full, classical, and quantum metric in the observable appear. If the dependence on $g_{\mu\nu}$ is non-linear, but analytic, the sum becomes a power series. This is especially relevant if the inverse metric appears. If it is non-analytic, then the sum becomes a term of unknown shape. Of course, there are always permutations where the classical and quantum metric are inserted at every possible place. The same formalism can also be applied to the vierbein instead, depending on circumstances.

If the leading term in (26) dominates, this is actually similar to what is called in slang "spontaneous electroweak symmetry breaking" in BEH physics, and could be called "spontaneous diffeomorphism breaking". The concept is the same. But likewise [4], this creates a wrong physical picture. The superficial "breaking" emerges from a two-step process. The first is to fix a gauge/coordinate system. This already breaks explicitly any symmetry. The second afterwards is then a split of the coordinate-fixed/gauge-fixed fields into a classical part (vacuum expectation value or classical metric) and a fluctuation field. As the split-off part is no longer invariant under arbitrary transformation, this can be considered as a manifestation of "breaking the symmetry" by the fixing. But it needs to be stressed that the actual breaking

---

[6]This choice is for simplicity.

[7]Note that this is fundamentally different from a background-field approach. The metric is split after gauge-fixing, and there is no separate symmetry transformations of either $g^c_{\mu\nu}$ or $\gamma_{\mu\nu}$. Only simultaneously transforming both in the same way is meaningful. A background metric in a background-field formalism would, instead, enjoy a full independent background diffeomorphism symmetry.

occurs when the gauge/coordinate system is fixed, not by the split. And it is this fixing which makes the split meaningful and possible. And that the split-off part is no longer symmetric under transformations is a consequence, not an origin. The non-trivial part is that dynamically the fluctuation field is small. Thus, it appears as if the physics is dominated by a quantity which does not posses the original symmetry, which gives rise to the idea that the symmetry is "spontaneously broken", as the most important contribution does not have the symmetry. But, in fact, it is really just the clever choice of gauge condition/coordinate systems which allows a split such that one part dominates. The symmetry is actually well and intact for any observable, though not for gauge-dependent/coordinate-dependent quantities. Because of that, the notion of "spontaneous breaking of the symmetry" will not be used here.

In fact, as will be seen there exists a split which shows how flat-space-time quantum-field theory emerges as a good approximation to physics. Because it turns out that in a suitable coordinate system splitting off $g^c_{\mu\nu} = \eta_{\mu\nu}$ yields for small $\gamma_{\mu\nu}$ flat-space-time quantum-field theory, and all the right properties for physics at, e. g., LHC. In this sense, experiment already tells us that such a split must be possible in (canonical) quantum gravity.

The avenues to this starts by noting that the expansion (26) permits an ordering in the size of the contributions by counting powers of $\gamma_{\mu\nu}$. This ordering can now be used to calculate physical quantities. In particular, if $O$ is a diffeomorphism-invariant quantity, then the right-hand sum must also be, even if the individual terms on the right-hand side are not. This shows in the present example immediately that the curvature is entirely given by the first term, and all quantum corrections to it vanish or cancel in the curvature gauge.

Applying this to the propagators of section 3, it is useful to first look at the argument, the invariant distance. Applying the expansion (26) yields

$$r = \min_{z(t)} \int_x^y dt\, g^c_{\mu\nu} \frac{dz^\mu(t)}{dt}\frac{dz^\nu(t)}{dt} + \left\langle \min_{z(t)} \int_x^y dt\, \gamma_{\mu\nu} \frac{dz^\mu(t)}{dt}\frac{dz^\nu(t)}{dt} \right\rangle = r^c + \rho. \qquad (27)$$

Thus, the invariant distance $r$ is the geodesic distance $r^c$ of the classical metric $g^c$ to which a quantum correction $\rho$ is added. This immediately gives also a test for the expansion. Only if $|\rho/r^c| \ll 1$, for $r^c$ not light-like, it can be expected to be a useful expansion. It should be noted that $\gamma_{\mu\nu}$ is definitely not small on individual configurations and can fluctuate locally wildly on individual configurations. The statement is essentially that all these fluctuations compensate on average. The first important result is that if $g^c_{\mu\nu} = \eta_{\mu\nu}$ this recovers that to leading order flat-space distances are the arguments on which propagators depend, just as in ordinary flat-space quantum-field theory. Especially, at leading order no expectation value needs to be performed anymore, as can be explicitly seen in (27).

Continue with $g^c_{\mu\nu} = \eta_{\mu\nu}$. In this case the curvature also vanishes already at leading order, recovering indeed flat space. As the operator $O_1$ (20) does not depend on the metric, its propagator reads

$$D_1(r) \;=\; \left\langle O_1(y)^\dagger O_1(x) \right\rangle(r) = \left\langle O_1(y)^\dagger O_1(x) \right\rangle(r^c + \rho) \qquad (28)$$

$$\;=\; \left\langle O_1(y)^\dagger O_1(x) \right\rangle(r^c) + \sum \partial_r^n D_1(r)|_{r^c}\, \rho^n + \delta(\rho), \qquad (29)$$

where $\delta$ collects all non-analytic contributions in $\rho$. The first term in (29) is the ordinary flat-space propagator. The second collects the quantum fluctuations of the metric on the distance between both events. Thus, as long as $\rho$ is indeed small compared to $r^c$, the remaining terms can be neglected, and the propagator is the one in the sense of quantum field theory. In the short-distance limit, however, it can be expected that $\rho$ at some point becomes comparable to $r^c$, and then quantum gravity effects affect the interpretation of the two-point function as a propagator, in the sense of quantum field theory.

So far, however, only the argument has been evaluated using the FMS expansion. The expectation value is still evaluated in a full quantum gravity setting, and contains the full

quantum fluctuations of the metric. It is now the second step of the FMS mechanism to apply a double expansion also to $\langle O_1(y)^\dagger O_1(x)\rangle(r^c)$, once in terms of the metric, and once in terms of all other coupling constants.

Because the operator $O_1$ (20) does not explicitly depend on the metric, this amounts to evaluate the expectation value in a power series in $\gamma$. Thus

$$D_1(r) = \left\langle O_1(y)^\dagger O_1(x)\right\rangle_{g^c_{\mu\nu}}(r^c) + \mathcal{O}(\gamma_{\mu\nu}). \tag{30}$$

But then the first term is just the ordinary propagator in the fixed metric $g^c_{\mu\nu}$. For $g^c_{\mu\nu} = \eta_{\mu\nu}$, this is the ordinary propagator of flat-space-time quantum field theory. In this sense, flat-space-time quantum field theory emerges as the leading term in the FMS expansion of quantum gravity.

In the standard model in flat space-time the propagator $D_1$ can be approximated by applying the FMS expansion a second time also to the Higgs field, yielding [1, 2, 4]

$$\phi = v + \Phi, \tag{31}$$

with $\langle\Phi\rangle = 0$ and $v$ the Higgs vacuum expectation value. This yields finally

$$D_1(r^c) = v^4 + v^2\langle\Phi(x)\Phi(y)\rangle + \mathcal{O}(v) = v^4 + v^2\langle\Phi(x)\Phi(y)\rangle_{\text{tl}} + \mathcal{O}(v, g, \lambda), \tag{32}$$

where the neglected terms only yield scattering thresholds. Especially, when expanding the term of order $v^2$ to lowest order in the particle physics coupling, this implies that $D_1$ is given by the tree-level Higgs propagator, and thus has exactly the same mass.

The final result (32) is thus the following statement: For values of the Newton coupling and cosmological constants where the average fluctuations of the full quantum metric around the classical metric is small, and over distances where geodesics are approximately the flat-space-time geodesics, and the parameters of particles physics yield small fluctuations around the respective vacuum expectation values, the full gauge-invariant, diffeomorphism invariant operator $O_1$ (20) behaves like the observed Higgs particle. This gives a fully physical leading-order description of the Higgs in quantum gravity, which agrees well with experiments. In this sense, the FMS mechanism explains how systematically flat-space-time quantum field theory emerges as a diffeomorphism-invariant limit of quantum gravity. This is a remarkable result: Starting from a manifestly diffeomorphism and gauge-invariant composite operator in full quantum gravity, it was systematically possible to calculate what mass in the scalar channel would be measured at the LHC: The one of the elementary Higgs.

This result can be systematically improved, by adding higher orders in the quantum corrections to the geodesics, the quantum fluctuations of the metric, and the particle physics fluctuations. However, this requires to suitably deal with ultraviolet problems both of particle physics and quantum gravity. As all effects seem to be small enough [4, 39] over distances relevant for CERN experiments, this does not spoil the agreement with experiment. Of course, evaluating the quantum gravity and standard model loop corrections requires an ultraviolet completion, which can, e. g., be due to asymptotic safety, including the matter sector [40].

There are many interesting directions how to augment the description of the Higgs with additional effects. One is clearly to go beyond the flat-space-time limit, both in the geodesic argument and in the classical expansion point. This yields quantum field theory in curved backgrounds [41, 42]. The other would be to introduce quantum gravity fluctuations at leading non-trivial order, e. g. in the context of asymptotic safety [26, 27]. Of course, conventional particle physics effects can be included as well [4].

The next step is to consider what an operator like $O_2$ (21) does, which includes explicitly the metric. This operator is, in fact, made up entirely of the metric and the spin connection. Thus, in the FMS expansion (26) the first term is the classical curvature scalar. If $g^c_{\mu\nu}$ is de

Sitter or in flat space time, this is just a constant, and in fact in both cases related to the cosmological constant, as expected in the scalar channel. Thus, the higher orders describe fluctuations on this dark energy background. Applying the FMS expansion yields, to lowest order, the expansion of $D_2$ as

$$D_2 = (6\Lambda)^2 + 3\Lambda \left\langle (g_c^{\mu\nu}\gamma_{\mu\nu}(x)) + x \leftrightarrow y \right\rangle \tag{33}$$

$$+ \left\langle (g_c^{\mu\nu}\gamma_{\mu\nu})(x)(g_c^{\rho\sigma}\gamma_{\rho\sigma})(y) \right\rangle (r_c) + \mathcal{O}(\gamma^2). \tag{34}$$

The second term vanishes, as there is no absolute space-time. The third term is the trace of the quantum metric, and describes thus a quantum excitation over the vacuum, which to leading order will depend again only on $r^c$. Thus, essentially this dilaton field creates scalar fluctuations around the dark energy background. This could be considered to be a gravity ball, or geon [21, 22].

Such a scalar particle, if reasonably stable and massive, is actually a dark matter candidate. Its properties can be calculated in various approximations [26, 27, 41–43], and will depend on the chosen classical metric. What happens under the assumption that it is indeed dark matter will be be explored more in section 6.

## 5 Particles with spin

To construct a manifestly invariant version of the $W$ and $Z$ bosons poses a fundamental problem. Because both particles need to be replaced by objects invariant under the weak gauge symmetry requires them to not be just the original gauge fields. The simplest operator to do this is actually the custodial current operator (12) [4]. However, this operator has no definite spin. Attempting to create an operator with a definite spin yields locally

$$J_a^u = e_a^\mu J_\mu^u. \tag{35}$$

Such an operator certainly transforms in any tangent space locally as desired. The corresponding correlator is then

$$D_{ab}^{uv} = \left\langle J_a^{u\dagger}(x) J_b^v(y) \right\rangle. \tag{36}$$

The argument is thus very similar to the one which requires the global custodial symmetry to create the observed (approximate) triplet of physical versions of the $W$ and $Z$ bosons [4]. The consequence of this is, as only the vierbein carries a tangent index, that the particles carrying spin considered here are effectively bound states of quantum gravity excitations and particle physics excitations. At least when considering the operator structure.

This yields now an challenging problem. As long as Lorentz transformations in the tangent space are local, it would be possible to perform them at both events independently. If averaging over such random transformations, an expression like (36) will vanish, for the same reasons as discussed in section 3. If the local Lorentz-symmetry would be gauged, like e. g. in Kibble-Sciama gravity[8] [18], the only possibility would be to add a dressing to (35), such that itself is invariant. But then no spin would remain, and thus no physically observable spin multiplets or selection rules. Similar considerations could apply to further generalizations of spin [19, 20].

The situation is, however, different. In canonical quantum gravity the local Lorentz transformations are only a reparametrization symmetry, and not part of a gauge symmetry. The

---

[8]It may be possible to consider spin in the same way as four-momentum or energy as something only creating an observable quantity in the low-energy/small curvature limit. Then one could alternatively also introduce, e. g., the spin connection as an additional dynamical gauge field [18], and perform a further FMS expansion on it, which will at leading (flat) order create spin as an effective global quantity, very much like four momentum. This avenue is not chosen here, because of the bias of the author to maintain spin as a global observable.

situation is therefore rather more similar to insist to evaluate quantities using two different coordinate systems in ordinary quantum field theory. Though such a theory is invariant under the choice of a coordinate system, it is still necessary to choose the same coordinate system to make meaningful statements.

A possibility is hence to choose the 'same' coordinate system in the tangent space at both events $x$ and $y$ in (36). How to achieve this in practice may be non-trivial. One possibility is to define the integration measure in (18) only modulo local Lorentz transformations, keeping global, i. e. event-independent, Lorentz transformations. This could also be obtained by either a 'gauge-fixing' term or by an actually dynamical term which breaks the local Lorentz symmetry explicitly to a global one. This would yield a preferred coordinate system in the arbitrary split in (1). Finally, an object of structure $\Sigma_{ab}(x,y)$, which transforms under local Lorentz transformations like $\Lambda^{-1}(y)\Sigma(x,y)\Lambda(x)$ and which reduces to $\eta_{ab}$ both in case $x \to y$ and in flat space, could be inserted into (36). This would be a kinematical analogue to the Wilson line in ordinary gauge theories. This last option is non-trivial, because it is ad hoc not clear whether such a purely kinematic object would not upset the notion of spin as a global quantity with measurable consequences. Still, it would be fundamentally different from a Wilson line as it does not involve a gauge symmetry, but only a reparametrization symmetry. The ultimate resolutions of this still requires further scrutiny.

However, for the purpose at hand, an FMS-style approach, any of these options will yield the same result at leading order. This can be seen as follows. Applying the full mechanism to the propagator (36) yields

$$
\begin{aligned}
D_{ab}^{uv} \quad &= \quad v^4 \left\langle \left( (e^c)_a^\mu W_\mu^u \right)(x) \left( (e^c)_b^\nu W_\nu^\nu \right)(y) \right\rangle (r^c) && (37)\\
&+ \quad v^4 \left\langle \left( \epsilon_a^\mu W_\mu^u \right)(x) \left( (e^c)_b^\nu W_b^\nu \right)(y) + x \leftrightarrow y \right\rangle && (38)\\
&+ \quad \left\langle \left( \epsilon_a^\mu W_\mu^u \right)(x) \left( \epsilon_b^\nu W_\nu^\nu \right)(y) \right\rangle + ... && (39)\\
&\overset{(e^c)_a^\mu = \delta_a^\mu}{=} \quad v^4 \left\langle W_a^u(x) W_b^\nu(y) \right\rangle (r^c) + ..., && (40)
\end{aligned}
$$

where the vierbein was split analogously as before into a classical part and the quantum part, $e_\mu^a = (e^c)_\mu^a + \epsilon_\mu^a$. This implies that the leading contribution is just the ordinary $W/Z$ propagator in flat space-time. Higher orders all contain already at least three fields, and are thus either scattering states or complicated bound states. Hence, in this controlled way also the $W/Z$ propagator emerge as the ordinary elementary particles in the intermediate distance regime. If a quantity like $\Sigma$ would have been introduced, it would become $\eta_{ab}$ in the last step, and thus yield the same result as using (36) directly.

The same applies to operators build entirely from the metric. Just like the $W$ and $Z$ gauge bosons cannot be the physical excitations in the standard model, so neither can the metric act like a physical object. It is necessary to construct a suitable fully diffeomorphism invariant quantity. The simplest suitable object is

$$
O_{ab} = e_a^\mu e_b^\nu R_{\mu\nu}, \qquad (41)
$$

i. e. the Ricci tensor projected into the tangent space, to make it a spin two particle in the same ways as the vector particle above. Of course, the same caveats apply here. Considering again the curvature gauge, the lowest non-vanishing order in the propagator is

$$
D_{abde} = \text{const.} + (e^c)_a^\mu (e^c)_b^\nu (e^c)_d^\rho (e^c)_e^\sigma \left\langle \gamma_{\mu\nu} \gamma_{\rho\sigma} \right\rangle + \mathcal{O}(\gamma^3). \qquad (42)
$$

As before, at intermediate distances this becomes the propagator of the quantum fluctuations of the metric around the gauge-fixed metric. At tree-level, this is a massless propagator, showing that the physical gravitational tensor particle is massless as well, consistent with the ex-

pectation. It should, however, be noted that this is a prediction about the very involved bound state operator (41) using the FMS mechanism[9].

Confirming this in any non-trivial calculation would validate the FMS expansion in quantum gravity, and would justify determining higher-order corrections. This would be especially interesting when it comes to the scalar excitation, as this would yield an interesting phenomenology, as the following, very speculative, section explores.

## 6 Speculative phenomenology

The previous setup incites a number of very interesting options. This section is entirely speculative, but indicates possible interesting directions where the ideas presented here could have consequences.

The most interesting option is the existence of a scalar particle-like fluctuation around the dark energy background, which could play the role of dark matter, the geon (34). There are two important ingredients to make it even a candidate.

One is that it is massive. At tree-level, this occurs because a non-vanishing cosmological constant induces such a tree-level mass, of order the cosmological constant [41, 42]. This makes the geon very light, but as a scalar, there is no stacking limit. Note that close to black holes the internal structure will be resolved, and usual arguments against such a very light dark matter candidate do not necessarily apply.

The other is that it is sufficiently stable. As it is very light, its decay channel is entirely into massless particles, i. e. photons and gravitons. Since there is no direct coupling to photons, only the graviton decay channel is interesting, which is mediated by the matrix element[10]

$$\mathcal{M} = \left\langle O_{ab} O^{ba} O_2 \right\rangle \approx \left\langle \gamma^{\mu\nu} \gamma_{\nu\mu} \gamma^\rho_\rho \right\rangle + \mathcal{O}(\gamma^4). \tag{43}$$

The second equality uses the FMS mechanism, to estimate the matrix element at intermediate distances. At tree-level, this matrix element is suppressed by the Newton coupling, and thus this decay is very weak. Hence, on relevant time scales, the geon could be stable.

The next question would then be its production in the early universe, as well as many other astrophysical constraints. However, this will happen potentially at strong quantum gravity fluctuations, and thus may require to go beyond the leading-order FMS mechanism.

The other phenomenological application to speculate about is what happens to black holes or other singularities. Obviously, this needs to be also recast into diffeomorphism-invariant operators. Considering a Schwarzschild black hole, this would be a scalar operator, i. e. such a black hole could actually even have overlap with (34). There are two options. Either there is a non-decomposable operator describing it, i. e. one which cannot be decomposed into separately diffeomorphism-invariant operators. Or it has overlap with a product of separately diffeomorphism-invariant operators. In this case, such a black hole would actually be akin to a neutron star, which is similarly described, but with baryon operators.

Thus, a black hole would then be rather a geon star, an object made up of many individual quantum particles. The properties of black holes, like the horizon, would then emerge as in-medium properties. The fact that, e. g., light cannot escape would then be very similar to how neutrinos cannot escape a forming neutron star. Just that the interactions would be mediated by strong gravitational interactions rather than by the weak interactions.

If a black hole is more complex, either by accumulating spin or matter, the corresponding description in terms of operators is again of the same kind. In fact, a black hole as observed in

---

[9]Note that massless composite particles with spin are already created similarly in conventional Yang-Mills theories [44–46].

[10]This is an observable process, and thus needs to be fully gauge invariant.

the cosmos, which has a sizable matter content, would then likely be a mix of interacting gauge-invariant particle and geon states, whose interactions become strong enough to collectively avoid large amounts of matter to escape. However, tunneling processes will still allow for some of these objects to escape, creating an alternative way of how Hawking radiation emerges. The evaporation of a black hole would then be merely a breaking apart of the object, very much like a neutron star would break apart if too much of its matter escapes by some process.

In this way, a black hole is not a very strange object at all. Rather, it is 'just' another form of how many particles behave. Note that such a picture of black holes is not that far off the picture obtained in so-called classicalization scenarios [47].

# 7 Summary

Herein, the FMS mechanism in quantum gravity is laid out, and its consequences of requiring manifest gauge invariance, both with respect to diffeomorphism invariance and quantum gauge symmetries in quantum gravity, are taken. This shows how, very like the situation in flat-space-time quantum field theory, quantum gravity states can be described in terms of the elementary excitations in suitable gauges. It also gives natural descriptions of objects, which behave like ordinary particles in space-times with negligible curvature. Finally, it addresses the necessities needed for spin to become an observable in the sense of a global symmetry.

While the FMS mechanism allows to estimate the behavior of quantum gravity at tree-level and in regions of weak curvature, this is not sufficient for calculations at loop level or at strong curvature, where the usual problems of quantum gravity will reappear. Here, the present setup needs to be supplemented by (weak) non-perturbative physics, e. g. asymptotic safety.

Of course, the present results are working under the assumption of a conventional form of quantum gravity. Replacing gravity with a different structure, e. g. string theory, may, or may not [48], yield different results. It is also not obvious what happens if supersymmetry is thrown into the mix. But using different dimensionalities, especially with additional compactified dimensions, additional fields, or a different action, will not qualitatively change anything of the presented structural results.

It should be noted that the present approach is related to loop quantum gravity in the same sense as ordinary gauge theory can be related to its formulation in terms of Wilson loops or other gauge-invariant variables. It would be equivalent on the level of observables, provided the same quantization would be performed, but utilizes the quasi-local formulation of gauge degrees of freedom at intermediate stages.

The options arising for dark matter and black holes as particle-like excitations of the gravitational fields, taking up the ideas of geons, is very interesting. Although, it would be somewhat depressing from the point of view of direct and indirect detection of dark matter. Still, this may yield potentially observable consequences for black hole-dark matter dynamics to be explored.

# Acknowledgments

I am grateful for many discussions on this subject to Reinhard Alkofer, Holger Gies, Astrid Eichhorn, Jan Pawlowski, Hèlios Sanchis-Alepuz, and Andreas Wipf and to Malcom Perry for his questions on the FMS mechanism and black holes, which initiated this research. I am grateful to Reinhard Alkofer and René Sondenheimer for a critical reading of the manuscript and valuable feedback.

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
