# Peer review of "The Fröhlich-Morchio-Strocchi mechanism and quantum gravity"

_SciPost Physics, doi:SciPost Phys. 8, 051 (2020)_

## Round 1 · Referee Report · Anonymous (Referee 1) · 2019-12-9

Report
The research ideas presented in the manuscript are highly original as they represent an unconventional new line of thought, transferring concepts from gauge theories of particle physics to quantum gravity. I am convinced that the ideas presented here will trigger some discussion within the quantum gravity community. Before I can recommend publication, I would appreciate a response by the author to the following set of questions:
Requested changes
(1) I find the discussion of local vs. global Lorentz symmetry slightly confusing: In the text, the author seems to consider global Lorentz transformations (which appear to be called "event-independent" transformations in the m/s). However, the explicit transformation rules, e.g., for the connection in the 2nd equation on p. 4, clearly exhibit that the formulation used by the author is locally Lorentz invariant (as it should). This distinction is particularly emphasized below Eq. (11), where the author points out the contradistinction between his preceding discussion and Kibble-Sciama gravity (which is locally Lorentz invariant). As this difference appears to be rather important (e.g., vanishing of correlators for spin?), a clarifying discussion may be needed.
(2) The author suggests an analogy between the vacuum expectation value of a scalar field in a theory with a Higgs mechanism and the split of the quantum metric into a classical metric and fluctuations. I wonder whether the analogy also implies that a classical background metric $g_{\mu\nu}\neq 0$ should also be viewed as a broken symmetry (in a gauge-fixed formulation)? Maybe the author could comment on this.
(3) The author advocates to use the expectation value of the invariant length (Eq. on p. 8) as the relevant argument for distance-type arguments in correlation functions. This seems to imply that a correlation function cannot be defined in terms of a single functional integral, since both expectation values (or $r$ and of the correlator) have to be computed. I wonder whether this should be viewed as a problem or a feature of quantum gravity as presented in the author's discussion.
(4) As may already be clear from my clumsy report, I suggest that all equations should be numbered.
Dear referee, I am grateful for the constructive and helpful report. I report here my answers and changes to the manuscript. I find the discussion of local vs. global Lorentz symmetry slightly confusing: In the text, the author seems to consider global Lorentz transformations (which appear to be called "event-independent" transformations in the m/s). However, the explicit transformation rules, e.g., for the connection in the 2nd equation on p. 4, clearly exhibit that the formulation used by the author is locally Lorentz invariant (as it should). This distinction is particularly emphasized below Eq. (11), where the author points out the contradistinction between his preceding discussion and Kibble-Sciama gravity (which is locally Lorentz invariant). As this difference appears to be rather important (e.g., vanishing of correlators for spin?), a clarifying discussion may be needed. The author suggests an analogy between the vacuum expectation value of a scalar field in a theory with a Higgs mechanism and the split of the quantum metric into a classical metric and fluctuations. I wonder whether the analogy also implies that a classical background metric gμν≠0 should also be viewed as a broken symmetry (in a gauge-fixed formulation)? Maybe the author could comment on this. The author advocates to use the expectation value of the invariant length (Eq. on p. 8) as the relevant argument for distance-type arguments in correlation functions. This seems to imply that a correlation function cannot be defined in terms of a single functional integral, since both expectation values (or r and of the correlator) have to be computed. I wonder whether this should be viewed as a problem or a feature of quantum gravity as presented in the author's discussion. --- @1: The referee is correct: There has been indeed a misunderstanding on my side, and as it was written, it was not adequate. The important step was to identify that the local Lorentz symmetry is not a gauge symmetry, but still a local reparametrization symmetry, in the sense that introducing the vierbein it allows at every event to do it differently. I have substantially expanded section 2 to discuss this issue, as it is indeed important for how to observe spin. In section 5, it is now discussed that it is necessary in order to have non-vanishing correlator with a non-trivial spin, this local reparametrization invariance needs to be taken into account. Three different options are discussed. It is emphasized that this is different from a gauge symmetry, as would be obtained when making the spin connection a dynamical field as proposed by Kibble-Sciama. As a fourth option also a second FMS-split is discussed in terms of a dynamical spin connection. In the end, I cannot (yet) completely solve the issue, and highlight this. This touches upon the question whether spin, like energy, should become a low-energy effective observable, or not. I argue for the former, but this is my personal bias. I hope this is now clearer, and conveys the message. I hope not being able to provide a final answer is acceptable at the current state of affairs. --- @2: In a background-field formulation, the background field enjoys full background symmetries, and thus I would not consider it as an analogy to the BEH mechanism. In the FMS approach presented, however, this is indeed different. But just as in the BEH case, the superficial breaking emerges from a two-step process: The first is to fix a gauge/coordinate system. This already breaks explicitly any such symmetry. The second afterwards is then a split of the coordinate/gauge-fixed fields into a classical part (vev or classical metric) and a fluctuation field. As the vev part is no longer invariant under arbitrary coordinate/gauge transformation, this can be considered as a manifestation of breaking the symmetry by the fixing. But as in the BEH case, I want to stress that the actual breaking occurs at the level of the gauge/coordinate system fixing, not by the split. And it is this fixing which makes the split meaningful and possible. And the split-off part is no longer symmetric under gauge/coordinate transformations. The non-trivial part is that dynamically the fluctuation field is small in the BEH case. Thus, it appears as if the physics is dominated by a quantity which does not posses the original symmetry, which gives rise to the idea of that the symmetry is spontaneously broken, as the most important contribution does not have the symmetry. But, in fact, it is really just the clever choice of gauge which allows a split such that one part dominates. The symmetry is actual well and intact for any observable, though not for gauge-dependent quantities. Provided also in quantum gravity such a split with a dominating part is possible, I would state that the same happens as in BEH physics, and, if you like, call this spontaneous breaking of diffeomorphism symmetry. However, personally I think this way of speaking is misleading, and I would prefer to avoid this. At any rate, I have added a corresponding discussion in the text between equations (23) and (24), and expanded footnote 5. --- @3: The correlation function itself is a single quantity. However, it depends on the events at which the fields are evaluated, and thus on objects, which are defined without a coordinate system. Especially,if the propagator is itself diffeomorphism invariant, this yields a diffeomorphism invariant result. The problem is thus to have a diffeomorphism invariant characterization of the two events. However, if the two events are not special in any way, which is assumed here by the quantization eq. (18), the only thing on which the propagator can depend are diffeomorphism-invariant characterizations of relative properties between both events. Still, the correlator is fully calculated at this point. It is just not particularly useful without somehow characterizing the two events on which it depends, if one wishes to, say, plot the propagator as a function of something. Of course, in intermediate steps on any fixed configuration the propagator can be calculated using coordinates just as usual. But this choice will become irrelevant when performing the path integral. In any given manifold, one such relative characterization between two events is their geodesic distance. However, when integrating over all manifolds, viz. metrics, this geodesic distance is also an expectation value, and thus to obtain this information, (21) is used. Thus, this is not needed for the correlator itself. But if one wants to think of the correlator not as an event-dependent quantity, but as a quantity depending on a diffeomorphism-invariant characterization of the relation between both events, this is necessary. And then this becomes a feature of quantum gravity, as was first, to my knowledge, remarked by Schaden in [23]. I do not think that this poses more than a (probably very involved) technical problem: Due to events being the underlying quantities on which the fields are defined, rather than on vectors in Minkowski space-time in QFT, this requires naturally a diffeomorphism-invariant characterization of this. It may well be that there is a better solution to this than eq. (21), but for the moment this appears a reasonable starting point. I have substantially expanded the discussion at the end of section 3 to include these points, as well as the notion of the fields to depend on events rather than vectors in the beginning of section 3. --- All equations are now numbered.
Author: Axel Maas on 2020-01-15 [id 709]
(in reply to Report 1 on 2019-12-09)Dear referee,
I am grateful for the constructive and helpful report. I report here my answers and changes to the manuscript.
I find the discussion of local vs. global Lorentz symmetry slightly confusing: In the text, the author seems to consider global Lorentz transformations (which appear to be called "event-independent" transformations in the m/s). However, the explicit transformation rules, e.g., for the connection in the 2nd equation on p. 4, clearly exhibit that the formulation used by the author is locally Lorentz invariant (as it should). This distinction is particularly emphasized below Eq. (11), where the author points out the contradistinction between his preceding discussion and Kibble-Sciama gravity (which is locally Lorentz invariant). As this difference appears to be rather important (e.g., vanishing of correlators for spin?), a clarifying discussion may be needed.
The author suggests an analogy between the vacuum expectation value of a scalar field in a theory with a Higgs mechanism and the split of the quantum metric into a classical metric and fluctuations. I wonder whether the analogy also implies that a classical background metric gμν≠0 should also be viewed as a broken symmetry (in a gauge-fixed formulation)? Maybe the author could comment on this.
The author advocates to use the expectation value of the invariant length (Eq. on p. 8) as the relevant argument for distance-type arguments in correlation functions. This seems to imply that a correlation function cannot be defined in terms of a single functional integral, since both expectation values (or r and of the correlator) have to be computed. I wonder whether this should be viewed as a problem or a feature of quantum gravity as presented in the author's discussion.
* * *
@1: The referee is correct: There has been indeed a misunderstanding on my side, and as it was written, it was not adequate.
The important step was to identify that the local Lorentz symmetry is not a gauge symmetry, but still a local reparametrization symmetry, in the sense that introducing the vierbein it allows at every event to do it differently. I have substantially expanded section 2 to discuss this issue, as it is indeed important for how to observe spin.
In section 5, it is now discussed that it is necessary in order to have non-vanishing correlator with a non-trivial spin, this local reparametrization invariance needs to be taken into account. Three different options are discussed. It is emphasized that this is different from a gauge symmetry, as would be obtained when making the spin connection a dynamical field as proposed by Kibble-Sciama. As a fourth option also a second FMS-split is discussed in terms of a dynamical spin connection.
In the end, I cannot (yet) completely solve the issue, and highlight this. This touches upon the question whether spin, like energy, should become a low-energy effective observable, or not. I argue for the former, but this is my personal bias. I hope this is now clearer, and conveys the message. I hope not being able to provide a final answer is acceptable at the current state of affairs.
* * *
@2: In a background-field formulation, the background field enjoys full background symmetries, and thus I would not consider it as an analogy to the BEH mechanism.
In the FMS approach presented, however, this is indeed different. But just as in the BEH case, the superficial breaking emerges from a two-step process: The first is to fix a gauge/coordinate system. This already breaks explicitly any such symmetry. The second afterwards is then a split of the coordinate/gauge-fixed fields into a classical part (vev or classical metric) and a fluctuation field. As the vev part is no longer invariant under arbitrary coordinate/gauge transformation, this can be considered as a manifestation of breaking the symmetry by the fixing.
But as in the BEH case, I want to stress that the actual breaking occurs at the level of the gauge/coordinate system fixing, not by the split. And it is this fixing which makes the split meaningful and possible. And the split-off part is no longer symmetric under gauge/coordinate transformations.
The non-trivial part is that dynamically the fluctuation field is small in the BEH case. Thus, it appears as if the physics is dominated by a quantity which does not posses the original symmetry, which gives rise to the idea of that the symmetry is spontaneously broken, as the most important contribution does not have the symmetry. But, in fact, it is really just the clever choice of gauge which allows a split such that one part dominates. The symmetry is actual well and intact for any observable, though not for gauge-dependent quantities.
Provided also in quantum gravity such a split with a dominating part is possible, I would state that the same happens as in BEH physics, and, if you like, call this spontaneous breaking of diffeomorphism symmetry. However, personally I think this way of speaking is misleading, and I would prefer to avoid this.
At any rate, I have added a corresponding discussion in the text between equations (23) and (24), and expanded footnote 5.
* * *
@3: The correlation function itself is a single quantity. However, it depends on the events at which the fields are evaluated, and thus on objects, which are defined without a coordinate system. Especially,if the propagator is itself diffeomorphism invariant, this yields a diffeomorphism invariant result. The problem is thus to have a diffeomorphism invariant characterization of the two events. However, if the two events are not special in any way, which is assumed here by the quantization eq. (18), the only thing on which the propagator can depend are diffeomorphism-invariant characterizations of relative properties between both events. Still, the correlator is fully calculated at this point. It is just not particularly useful without somehow characterizing the two events on which it depends, if one wishes to, say, plot the propagator as a function of something. Of course, in intermediate steps on any fixed configuration the propagator can be calculated using coordinates just as usual. But this choice will become irrelevant when performing the path integral.
In any given manifold, one such relative characterization between two events is their geodesic distance. However, when integrating over all manifolds, viz. metrics, this geodesic distance is also an expectation value, and thus to obtain this information, (21) is used. Thus, this is not needed for the correlator itself. But if one wants to think of the correlator not as an event-dependent quantity, but as a quantity depending on a diffeomorphism-invariant characterization of the relation between both events, this is necessary.
And then this becomes a feature of quantum gravity, as was first, to my knowledge, remarked by Schaden in [23].
I do not think that this poses more than a (probably very involved) technical problem: Due to events being the underlying quantities on which the fields are defined, rather than on vectors in Minkowski space-time in QFT, this requires naturally a diffeomorphism-invariant characterization of this. It may well be that there is a better solution to this than eq. (21), but for the moment this appears a reasonable starting point.
I have substantially expanded the discussion at the end of section 3 to include these points, as well as the notion of the fields to depend on events rather than vectors in the beginning of section 3.
* * *
All equations are now numbered.

---

## Round 2 · Referee Report · Anonymous (Referee 2) · 2020-1-18

Strengths

1- The work applies ideas originating from particle physics in the context of gravity. This results in a highly innovative (and partially speculative) new perspective which will certainly trigger interesting discussions in the field.

2- The construction of observables in the context of quantum gravity is a highly non-trivial problem, see e.g. arXiv:1507.07921. The work makes interesting progress in this direction.

3- The article outlines a potential connection between fundamental gravity research and a phenomenologically interesting dark matter candidate, the “geon”.

Weaknesses

1- In some places, the work is short on details. In particular the discussion about the status of a vanishing vacuum expectation value for the spacetime metric could have taken more room.

Report

I find the work interesting and rich of original ideas. It certainly deserves publication in SciPost Physics.

Requested changes

I leave it up to the author whether he still would like to comment on and suitably implement the following suggestions:

1- The vanishing of the vacuum expectation value for the spacetime metric is an interesting statement. The arguments leading to this conclusion are clear. There are two potentially far-reaching consequences where it would be interesting to learn the author’s opinion. Firstly, does the author suggest that one should focus on directly computing expectation values of the physical length $<ds^2> = < g_{\mu\nu} dx^\mu dx^\nu >$ without referring to the concept of a vacuum expectation value for $g_{\mu\nu}$? Secondly, it is commonly accepted that one could compute a non-vanishing expectation value $<g_{\mu\nu}>$ by solving the equations of motion derived from the effective action of gravity. At this stage, it is not clear to me how to reconcile these two pictures, so it may be worthwhile to clarify this connection.

2- The role of eq. (14) is not clear. In principle $J_\mu^u$ is independent of the spin-connection. The equivalence principle would nevertheless suggest that the conservation law should be constructed from a covariant derivative including the Levi-Civita connection. Since in general $\partial_\mu$ will not commute with the metric $\partial_\mu J^\mu \not = \partial^\mu J_\mu$ which seems odd.

3- There are a few stylistic remarks which would improve the readability of the work. E.g., the definition of the operators $O_1$ and $O_2$ could be put into an equation and then referred back to when their two-point-functions are studied. The paragraph containing eq. (23) provided a stumbling block. My first reaction was that this cannot be true in general (the remaining paragraph explains this in detail). So perhaps one could describe the generic situation first and then use eq. (23) as a particular example.

  • validity: high
  • significance: high
  • originality: high
  • clarity: good
  • formatting: good
  • grammar: excellent

Author:  Axel Maas  on 2020-02-04  [id 725]

(in reply to Report 1 on 2020-01-18)
Category:
remark
answer to question
correction

Dear referee,

thank you for the kind report, and leaving me the option, whether I want to answer. However, the editor has decided that I have to address the issues, so my own decision is not necessary.

In response to your questions, I have the following answers and made the following changes:

1- The vanishing of the vacuum expectation value for the spacetime metric is an interesting statement. The arguments leading to this conclusion are clear. There are two potentially far-reaching consequences where it would be interesting to learn the author’s opinion. Firstly, does the author suggest that one should focus on directly computing expectation values of the physical length <ds^2>=<g_μν dx_μdx_ν> without referring to the concept of a vacuum expectation value for g_μν?

Yes and no. If a gauge and coordinate system is fixed, one can talk also about the metric itself. However, just as with other quantum fields, I expect still that any expectation value <g(x)> can still either only vanish if no event is special (as long as the gauge condition treats all event equally), or as an alternative would be a metric, which is essentially x-independent, i.e. an isotropic and homogeneous one. Though to express it also a coordinate system needs to be fixed. Thus, rather correlation functions <g(x)g(y)> would make more sense.

Without fixing the gauge a similar statement would apply to <ds^2> - as long as it is evaluated at fixed events, and with an indefinite metric, I expect this to average to zero. Rather, histograms over all events would be interesting, giving average distributions of such quantities. Alternatively, integrated curvature would probably be an alternative to characterize the average geometry of space-time.

I have added these statements in section 3 when discussing expectation values.

Secondly, it is commonly accepted that one could compute a non-vanishing expectation value <g_μν> by solving the equations of motion derived from the effective action of gravity. At this stage, it is not clear to me how to reconcile these two pictures, so it may be worthwhile to clarify this connection.

If the quantum effective action is gauge-fixed and a coordinate system is chosen, the same should apply as before, and one can get a non-vanishing <g_mn>. If it is not gauge-fixed, I would expect that only zero would be such a solution. But in the latter case, a diffeomorphism-invariant formulation of the quantum effective action would likely be qualitatively similar to having Yang-Mills theory formulated in Wilson loops, and then as little can be said about the metric as can be said about the gauge fields in such a formulation of Yang-Mills theory. However, I am not sure how to formulate such a quantum effective action of either Yang-Mills theory or gravity at this stage, and can thus not provide any substantial support for my expectation.

At any rate, as before, if the quantum effective action treats all events equal, so should any such expectation value of the metric. Hence, I expect for such solutions the same as said above.

Of course, if for some reason events become special, anything can happen.

2- The role of eq. (14) is not clear. In principle J^u_μ is independent of the spin-connection. The equivalence principle would nevertheless suggest that the conservation law should be constructed from a covariant derivative including the Levi-Civita connection. Since in general ∂_μ will not commute with the metric ∂_μ J^μ≠∂^μ J_μ which seems odd.

The argument would be that J_mu is actually an expression, which involves only the scalar fields and the covariant derivative Delta_mu. On each of these D_mu should act only like partial_mu, when decomposing it into its elements. This would be like the electromagnetic current (see [18]), and comes about because the charge is actually carried by scalars, which do not involve a non-trivial coupling to the metric. I have added a corresponding statement. This could also be seen by explicitly expanding all terms, and executing the covariant derivatives explicitly. In this case, and only this case, because pd_mu J^mu=0 and pd^mu J_mu=0, there is no contradiction.

3- There are a few stylistic remarks which would improve the readability of the work. E.g., the definition of the operators O_1 and O_2 could be put into an equation and then referred back to when their two-point-functions are studied. The paragraph containing eq. (23) provided a stumbling block. My first reaction was that this cannot be true in general (the remaining paragraph explains this in detail). So perhaps one could describe the generic situation first and then use eq. (23) as a particular example.

I have followed the recommendations. I have also added a citation to the mentioned article, which I found very useful.

---

## Editorial Decision

published